# Macrophages self-generate and refine chemotactic gradients during migration towards complement C5a

Abhimanyu Kiran[1,2], Peter A. Thomason[1], David M. Versluis[2], Peggy I. Paschke[1], Hannah Donnelly[1¤], Luke Tweedy[1], Sergio Lilla[1], Isabel Bravo-Ferrer[2], Amy Shergold[1], Ryan Devlin[1], Edward W. Roberts[1], Robert H. Insall [1,2]*

1 CRUK Scotland Institute, Bearsden, Glasgow, United Kingdom, 2 Division of Cell & Developmental Biology, University College London, London, United Kingdom,

¤ Current address: School of Molecular Biosciences, University of Glasgow, Glasgow, United Kingdom
* R.Insall@ucl.ac.uk

## Abstract

Macrophages rely on efficient chemotaxis to locate sites of infection and tissue damage. One general strategy that enhances chemotactic accuracy is the use of self-generated gradients, where cells locally deplete attractants to create or sharpen guidance cues. Here, we show that mouse bone marrow–derived macrophages (BMDMs) migrate toward the complement component C5a using this strategy. Cells actively deplete C5a from their surroundings, establishing local gradients as fresh attractant diffuses inward. We visualized this process in real time with fluorescent C5a and reproduced its dynamics using computational models. C5a depletion is mediated primarily by C5aR1-dependent endocytosis. This mechanism produces complex responses, with different C5a concentrations inducing temporally distinct waves of migration, and maximal chemotaxis occurring below 10 nM C5a. As expected, increasing C5a concentrations recruit more cells. In contrast, human macrophages inactivate C5a mainly through carboxypeptidase-mediated degradation, yielding a higher optimal concentration (~30 nM) and distinct migratory dynamics. Both species also deplete externally-imposed C5a gradients, sharpening them and enhancing guidance. These findings identify C5a degradation as a critical mechanism by which macrophages extract directional information from their environment. Self-generated gradient formation, despite different mechanisms across species, emerges as a conserved and versatile strategy for immune navigation.

## Introduction

Cell migration is a fundamental biological process involved in embryo implantation, development, cancer metastasis, wound healing, and immune cell function [1,2]. While cells can migrate randomly, it is extremely inefficient over most physiological

**Data availability statement:** Raw data are additionally available at the UCL archive: https://doi.org/10.5522/04/31211095, and the code is available at https://github.com/DMvers/Chem-O-Taxis_macrophage_c5a and archived at https://doi.org/10.5281/zenodo.18882803.

**Funding:** This work was supported by the Wellcome Trust (https://wellcome.org; grant 221786/Z/20/Z to RHI), UK Medical Research Council (https://www.ukri.org/councils/mrc/; grant MR/X000702/1 to RHI) and Cancer Research UK (https://www.cancerresearchuk.org; grant A31287 to the CRUK Scotland Institute and A1920 to EWR). The funders had no role in study design, data collection and analysis, decision to publish, or preparation of the manuscript.

**Abbreviations:** BMDMs, bone marrow–derived macrophages; CPM, carboxypeptidase M; GPCR, G-protein coupled receptor; LPS, lipopolysaccharide; mCSF, macrophage colony stimulating factor; PRRs, pattern recognition receptors; SEM, standard error of the mean; SGG, self-generated gradient; TLR4, toll-like receptor 4.

scales; efficient migration in vivo is usually guided by directional cues from the environment. Among the best-understood guidance cues are diffusible chemicals—chemoattractants—which form gradients interpreted by receptors on the cell surface. The direction and steepness of the gradients provide steering cues that are interpreted through receptors, most often seven transmembrane receptors that couple to G proteins, yielding chemotaxis-cell migration directed towards the source of the chemoattractant [3].

For immune cells, chemoattractants fall into two main categories: those produced by the target (e.g., formyl peptides like fMLP from bacteria) and those produced by the host, such as chemokines or complement-derived peptides like C5a [4,5]. Immune cells detect these gradients and efficiently migrate toward their sources [6–8].

Macrophages are a critical immune cell population, essential to the body's defence system, that respond rapidly to infections. They perform diverse functions including pathogen engulfment (phagocytosis), antigen presentation, and tissue repair. Macrophages can arise from circulating monocytes that infiltrate tissues, or from long-lived tissue-resident populations established during early development [9,10].

Within tissues, macrophages constantly sample their environment using pattern recognition receptors (PRRs) to detect pathogen-associated molecular patterns. Once activated, they migrate directionally toward infection sites, guided by chemokines from other immune cells and complement factors such as C5a and C3a [11,12]. C5a is one of the most potent chemoattractants known for macrophages and also neutrophils. It is produced at sites of complement activation, a clear hallmark of tissue damage, by proteolysis from the much larger C5 molecule.

However, the gradients of these chemical signals at infection sites are not always well-defined. Attractants are often initially uniform or spread into shallow gradients by tissues. Consequently, to navigate effectively, cells must locally modify or degrade these constant concentrations to create steep gradients, facilitating their precise migration towards the source of infection. This phenomenon, known as self-generated chemotaxis, has been observed in various cell types, including various cancers, dendritic cells, and model organisms such as zebrafish and *Dictyostelium* [13–16].

Although C5a is known as a dominant chemoattractant for macrophages, it is unclear how macrophages navigate when C5a is abundant but not spatially structured. We therefore asked whether they, like other systems, could generate their own navigational cues. C5a biology makes it an especially suitable context to test self-generated guidance. It is generated rapidly and diffusely during complement activation [4], often creating high but poorly directional concentrations [17]. Macrophages express high levels of its receptor, C5aR1. At the same time, C5a is inactivated by multiple macrophage-expressed mechanisms, including endocytosis, protease-mediated degradation, and enzymatic trimming. These features suggest that macrophages are not merely passive responders but may actively remodel C5a fields to create navigational cues.

We now demonstrate that mouse bone marrow-derived macrophages (BMDMs) employ self-generated gradients to steer toward C5a. The cells endocytose or

degrade the C5a ligand while they respond to it, creating gradients even in initially uniform environments and sharpening preexisting gradients around the responders. This represents, to our knowledge, the first demonstration of self-generated chemotaxis in macrophages, and fills a missing piece in our picture of host defence.

## Results

### Macrophages create and migrate up chemotactic gradients of C5a

Complement C5a is among the most potent chemoattractants for macrophages [18]. In most experimental setups, it is presented to cells as a premade linear gradient, typically formed between two reservoirs, one containing no C5a and the other a substantial concentration. However, our research on cells including different cancer types and *Dictyostelium*, backed up by computational models, has shown that cells can chemotax more efficiently to gradients formed locally by the responding cells themselves [19–21]. We therefore tested the ability of BMDMs to self-generate gradients of C5a and navigate using them.

We filled an Insall chemotaxis chamber (Fig 1A) [22] with a homogeneous concentration of 10 nM C5a (that is, initially 10 nM C5a in both wells and the bridge between) and introduced $1.5 \times 10^6$ BMDMs/ml to the outer well. If the cells efficiently depleted the C5a present in the outer well, a chemotactic gradient would be established along the bridge; if they did not, the cells would migrate randomly without steering cues, and mostly remain in the well. Fig 1B and S1 Movie show efficient self-generated chemotaxis of remarkable efficiency—despite initially homogeneous C5a, macrophages oriented accurately towards the opposite well and migrated directly away from their initial reservoir. The negative control showed random migration over this timescale (Fig 1B and S1 Movie), confirming that the migration depends on the presence of C5a. Thus, macrophages are able to self-generate C5a gradients and simultaneously respond to them.

Inspired by the experimental observations, a hybrid agent-based computational simulation was performed, based on earlier modeling work by our lab [15] and using parameters derived from our measurements, listed in S1 Table. In these simulations, each cell independently degrades the chemoattractant in its immediate vicinity and moves up gradients of attractant. When initialized with a uniform concentration of chemoattractant, simulations were able to replicate the experimental observations (Fig 1C and S2 Movie), indicating that locally degrading attractant can lead to an effective self-generated gradient, and effectively illustrating the mechanism underlying the chemotaxis.

One defining feature of self-generated chemotactic gradients is a collective wave at the front of the population of migrating cells [23]. Macrophages migrating in C5a show a particularly clear front wave (Fig 1B at 18 h; red box in S1 Fig). These cells are particularly accurately aligned towards the opposite well and migrate as a coherent group. Quantification of this collective response to an initially uniform 10 nM C5a is shown in Fig 1D. Another behavior particularly noted in self-generated chemotaxis is that high attractant concentrations cause a delay before the wave of chemotaxis begins (the time it takes for the cells to clear the attractant out of their well) and slower directional migration of the cells in the wave [23]. We analyzed the response of macrophages to a substantially higher concentration of C5a (100 nM)., S1 Fig, and S3 Movie show that the cells took significantly longer to start chemotaxing and enter the bridge than at 10 nM, though still obviously better than the negative control. Interestingly, the alignment of the cells in the front wave was strikingly different in both cases. With 10 nM C5a, the cells in the front wave were inclined to ~90° to the bridge (S1 Fig); with 100 nM, the alignment was nearer ~45°, implying a much less legible gradient, presumably because cells struggled to clear the C5a from their rears, or because persistent migration carried them into regions of C5a saturation.

It is also clear that the positive control with a simple 0–10 nM gradient (Fig 1B, S4 Movie) shows a wave of particularly well-oriented cells at the front, with decreased chemotaxis behind. This behavior contrasts with the continuous response expected from cells responding simply to an imposed gradient. This implies that the same mechanisms that create self-generated gradients also act to shape imposed gradients, as has been seen, for example, in melanoma chemotaxis [19]. The formation and propagation of the cell wave is quantified in Fig 1D. Simulations of cells in initial gradient of C5a

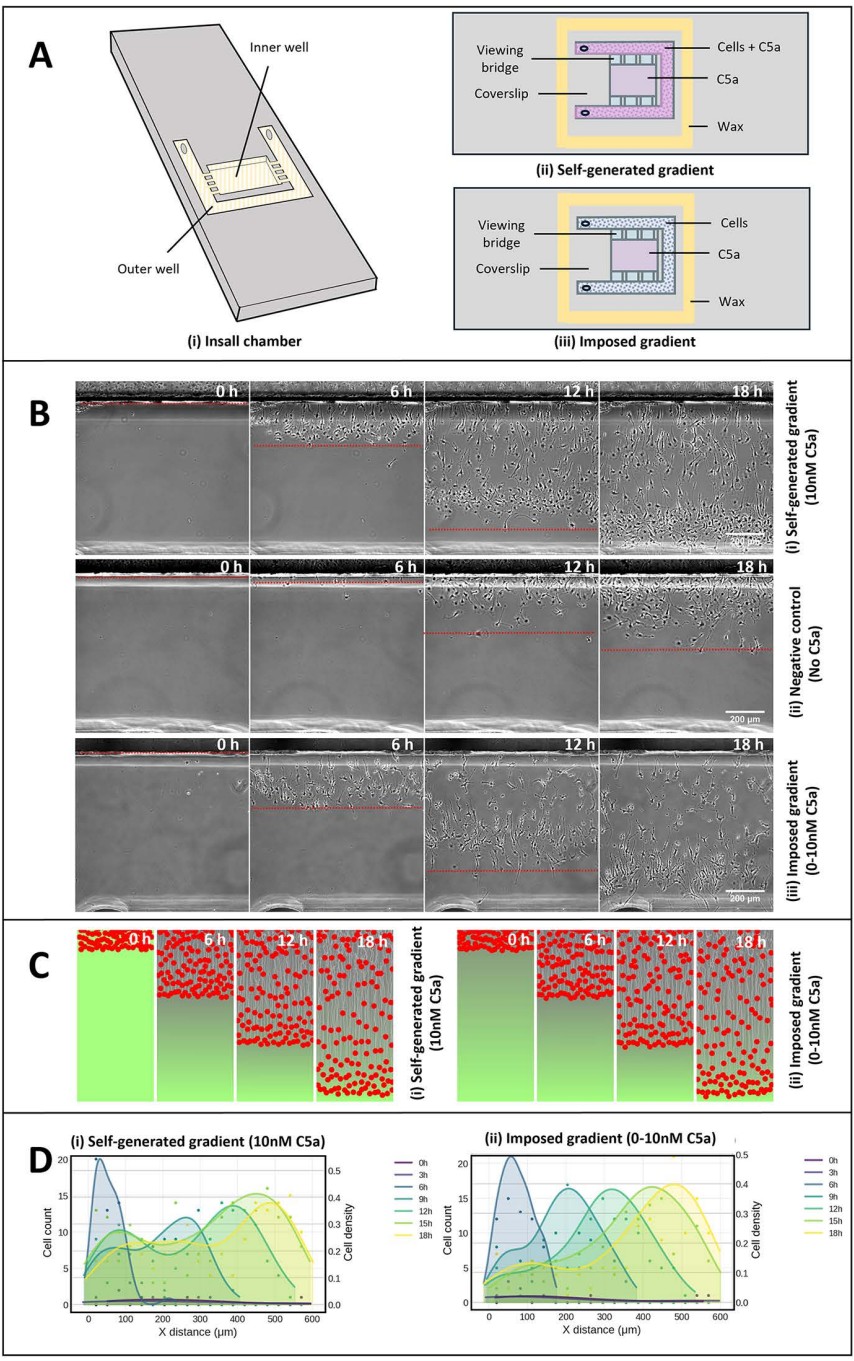

**Fig 1. Self-generated chemotaxis of mouse macrophages to C5a.** (A) Schematic of (i) Insall chamber geometry, (ii) self-generated gradient assay, and (iii) imposed gradient assay. (B) Macrophage responses to (i) an initially homogeneous concentration of 10 nM C5a, (ii) negative control (no C5a), and (iii) an imposed gradient of 0–10 nM C5a. (C) Computational simulation, using parameters measured from experimental data, showing (i) cells depleting C5a from a homogeneous 10 nM environment and migrating up the resulting self-generated gradient, and (ii) cells removing the C5a from an imposed gradient of 0–10 nM and migrating up it. (D) Quantification with smoothed curves obtained using kernel-density estimation with Gaussian kernels of front waves formed by cells in response to (i) an initially homogeneous concentration of 10 nM C5a, and (ii) an imposed gradient of 0–10 nM C5a. Original data underlying this figure can be found in https://doi.org/10.5522/04/31211095, and code at https://doi.org/10.5281/zenodo.18882803.

were able to reproduce the experimental observations in the same way they recapitulated self-generated gradients (Fig 1C).

## Effects of lipopolysaccharide activation

Activation in the presence of bacterial or pathogen traces is a fundamental feature of macrophage biology. We therefore examined self-generated chemotaxis in LPS- activated and inactivated macrophages. The LPS binds and activates the Toll-like receptor 4 (TLR4) on the surface of the macrophage, triggering a signaling cascade that alters gene expression and leads to the production of inflammatory cytokines and mediators [24].

LPS stimulation was not essential for self-generated chemotaxis. Both activated and inactivated macrophages respond effectively to homogeneous C5a, and start to migrate at the same time. However, the inactivated macrophages chemotax less accurately than LPS-activated ones. Clearly, both activated and inactivated macrophages express a high enough level of C5a receptors to steer. The differences, therefore, more likely reflect changes in the pathways that connect gradients to migration and steering, or proinflammatory responses after activation (Fig 2).

## Dose dependence of self-generated C5a chemotaxis

In other cell types, we have shown that the number of cells in the front wave of a self-generated gradient is approximately proportional to the attractant concentration [15,21]. We tested whether this holds for BMDMs using self-generated

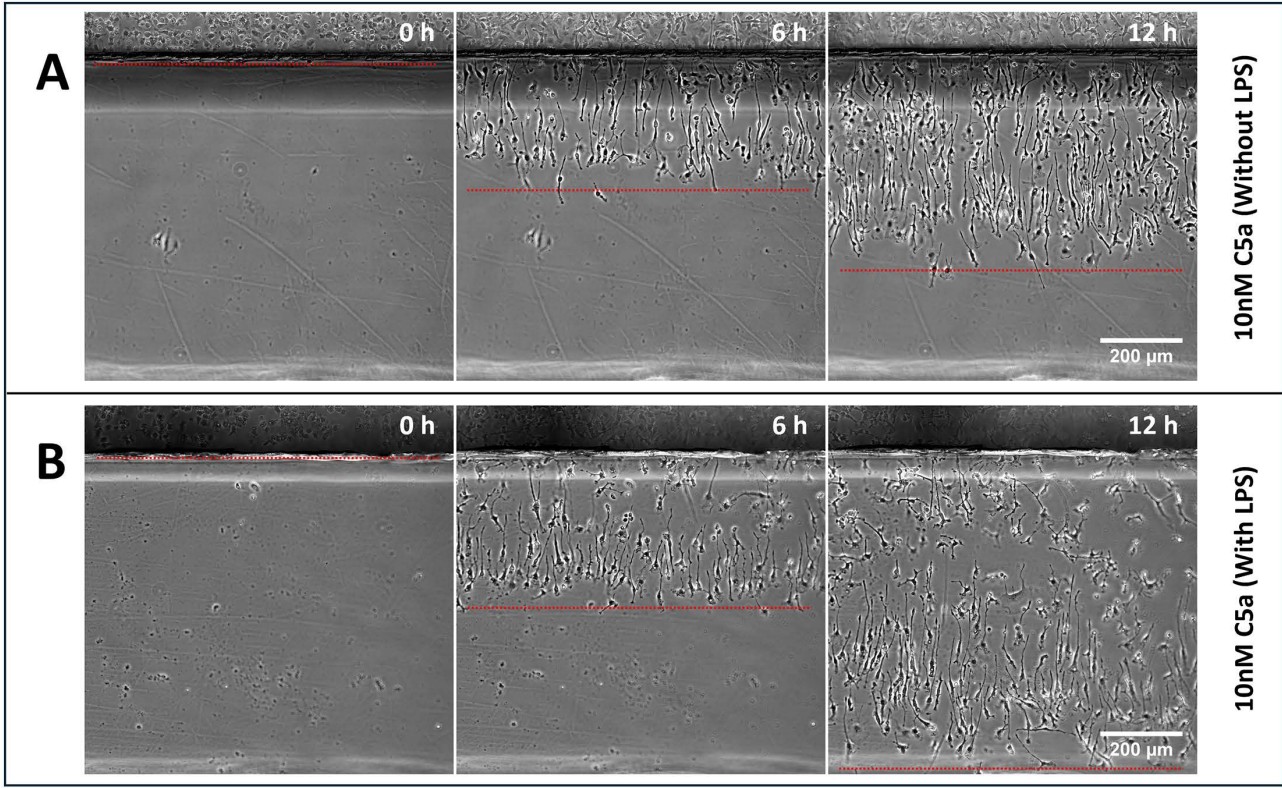

**Fig 2. LPS stimulation of self-generated chemotaxis of BMDMs to C5a.** BMDM migration in response to homogeneous C5a in the absence (A) or presence (B) of lipopolysaccharide (LPS) stimulation. In both conditions, cells established self-generated gradients and migrated across the bridge, but LPS-activated macrophages showed greater accuracy and coherence in their directional movement. Red dashed lines outline the regions occupied by migrating cells on the bridge.

gradients formed from initial C5a concentrations of 0.3 nM, 3 nM, and 30 nM (S5 Movie). Figure 3A shows responses of the same initial cell density after 6 h. As expected, few cells entered the bridge at 0.3 nM C5a, because that is all that is required to inactivate or remove all of the C5a as it diffuses from the source well. With 3 nM C5a, the number of cells in the front wave was clearly greater (Fig 3A), but only by about 2-fold despite the 10× increase in chemoattractant. The same was true at the yet higher concentration of 30 nM C5a; after an initial delay for the cells to deplete the pool (see above), a larger wave still was formed (Fig 3A), but again with less than 3-fold further increase in cell number despite a 10× increase in attractant. Again, negative controls with no C5a gave negligible migration under these conditions (Fig 3B). Note also the slower response at higher concentrations as described earlier.

We would expect an approximately linear relationship between attractant concentration and the number of cells in the wave. The actual result suggests an extra level of complexity. One possibility is that cells are increasing their rate of breakdown at higher C5a concentrations, for example, proposed induction of extra breakdown enzymes at high ligand occupancy as a mechanism to increase robustness [21]. Another is that cells are communicating with one another through additional mechanisms than simple C5a depletion. To follow this, we examined hybrid agent-based computational simulations. These were sufficient to recreate the experimentally observed dose responses (S6 Movie).

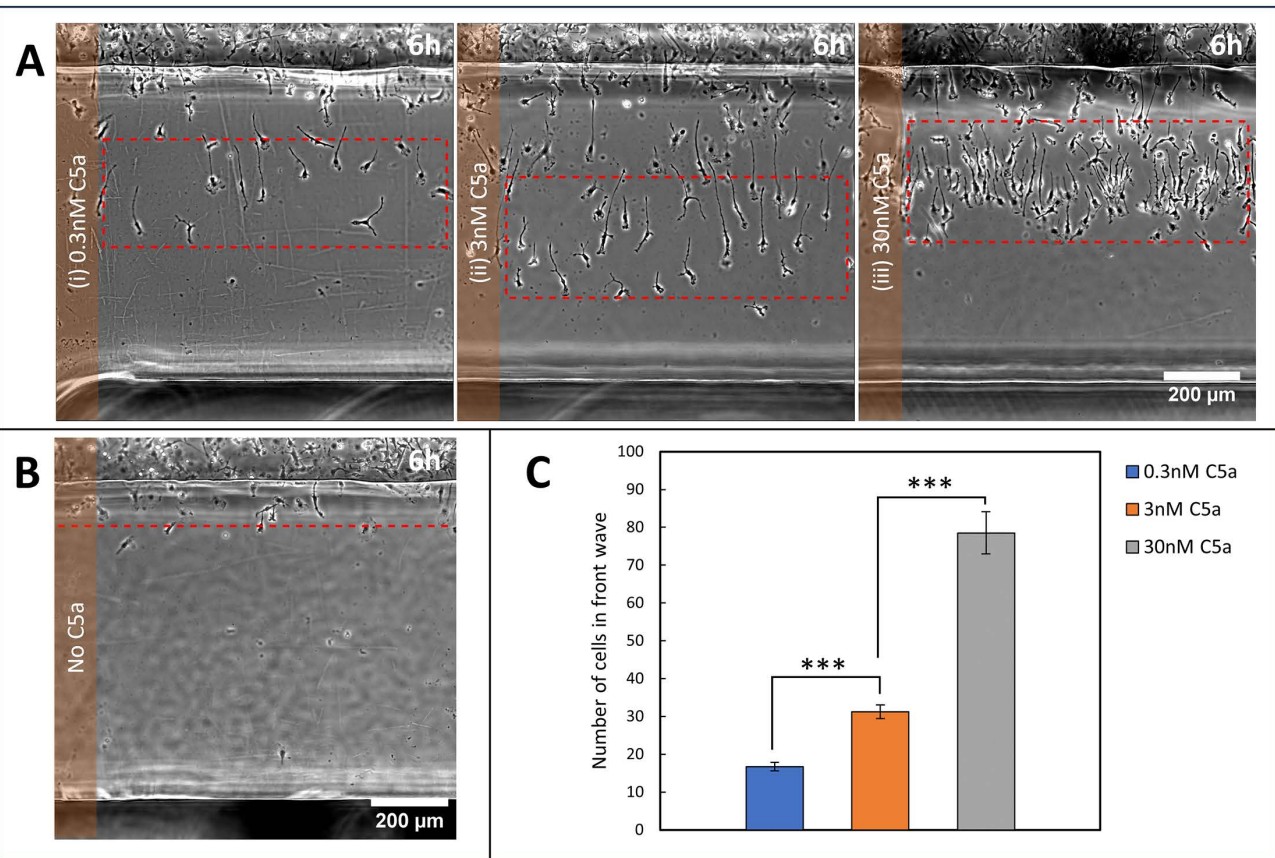

**Fig 3. Dose dependence of self-generated chemotaxis to C5a.** (A) Representative images showing BMDM migration after 6 h in initially homogeneous C5a concentrations of (i) 0.3 nM, (ii) 3 nM, and (iii) 30 nM. Red dashed boxes highlight the front wave of cells. (B) Negative control (no C5a), showing negligible migration. The red dashed line indicates the region assessed for cell migration. (C) Quantification of cells in the front wave at different C5a concentrations after 6 h. Data are shown as mean±SE. Statistical significance was assessed using Student *t* test; *** indicates *p* < 0.0001. The data underlying this figure can be found in https://doi.org/10.5522/04/31211095.

We also examined the effect of changes in initial cell density. These supported earlier observations that the number of cells in the front wave depends only on the attractant concentration [15]; cell density only affects the time before the wave is initialized (compare the time required for cells to migrate upon the bridge for the same C5a concentration in S2A, S2B, and S2C Fig).

## BMDMs deplete C5a from their surroundings using endocytosis

To perform self-generated chemotaxis, macrophages must get rid of the C5a in their local environment to form locally steep gradients. Two general mechanisms for this have been described—first, using enzymes that convert the chemo-attractant into an inactive form [19,25], and second, by receptor-mediated endocytosis of the chemoattractant [13,26]. The enzymes may be membrane-bound with their catalytic domains exposed to the extracellular medium, like LPP3, or secreted, like phosphodiesterases. One strong candidate enzyme, carboxypeptidase M, is known to inactivate C5a in a complicated fashion by trimming off the C-terminal arginine [27], and is specifically expressed in human macrophages [28,29]. However, mouse macrophages such as our BMDMs do not express carboxypeptidase M (see, for example, the Immunological Genome Project–https://rstats.immgen.org/Skyline/skyline.html). We also examined C5a by mass spectrometry after incubation with BMDMs. The trimmed C5a (C5a-des-R) was not detected when mouse cells were incubated with C5a, whereas it was detected in all samples using THP-1 cultured human cells (cultured cells with macrophage-like gene expression and behavior, differentiated from a monocytic leukaemia line, as primary human cells were unavailable; S3 Fig). Interestingly, a hitherto undescribed double-trimmed form (C5a-des-GR) was also seen in the human and not mouse experiments. Mouse macrophages express a range of other secreted and membrane-bound proteases, but CPM is the only enzyme known to degrade C5a.

To further distinguish endocytic from enzymatic degradation, we used commercially available C5a labeled with fluorescent AF647 at its N-terminus. Extracellular enzymes such as proteases do not change the fluorescence of this molecule when they attack it—trimming the C5a into an inactive form does not alter its fluorescence, so active and inactive molecules fluoresce the same. However, endocytosis should be visible as a loss of fluorescence from the medium and a sharp gain in vesicles inside the cell [30]. We visualized both medium and intracellular detail in BMDMs migrating up a self-generated gradient of fluorescently labeled C5a-AF647, examining both using a confocal microscope, after the cells had moved some distance across the bridge. The results are striking (Fig 4)—BMDMs internalize the labeled C5a, and remove it from the medium, creating a gradient. The chambers, which were set up with homogenous C5a, show a clear gradient after a few hours' incubation with cells—low in the immediate vicinity of the BMDMs, becoming higher by 200 µm away. Soon after the chamber has been assembled, the macrophages contain punctate fluorescence, reflecting the endocytosed C5a and its breakdown products.

Thus, BMDMs set up their own chemoattractant gradients by endocytosing C5a from their surroundings, and retaining it, presumably for lysosomal breakdown.

Endocytosed C5a could be recycled and re-secreted or targeted to lysosomes and broken down. To explore this mechanism, cells were seeded in Petri dishes at a similar density to the chemotaxis chambers, exposed to 5 nM of C5a-AF647 for 1 h, then washed with fresh medium and examined after different intervals (Fig 4C). This clearly shows the cells retain the C5a they take up. The fluorescent signal appeared to be in endocytic vesicles and eventually localized in the perinuclear region (Fig 4C), and was not released in significant quantities over three hours.

## Comparison between human and mouse macrophages

We hypothesized that the presence of CPM in human macrophages might alter the way they made and responded to self-generated gradients. Human macrophages could use the surface bound enzyme to get rid of C5a in their local environment, in contrast to the receptor-mediated endocytosis employed by the mouse macrophages. We tested this hypothesis several ways. First, we observed the change in the extracellular imposed gradient of fluorescent C5a-AF647 by the

PLOS Biology

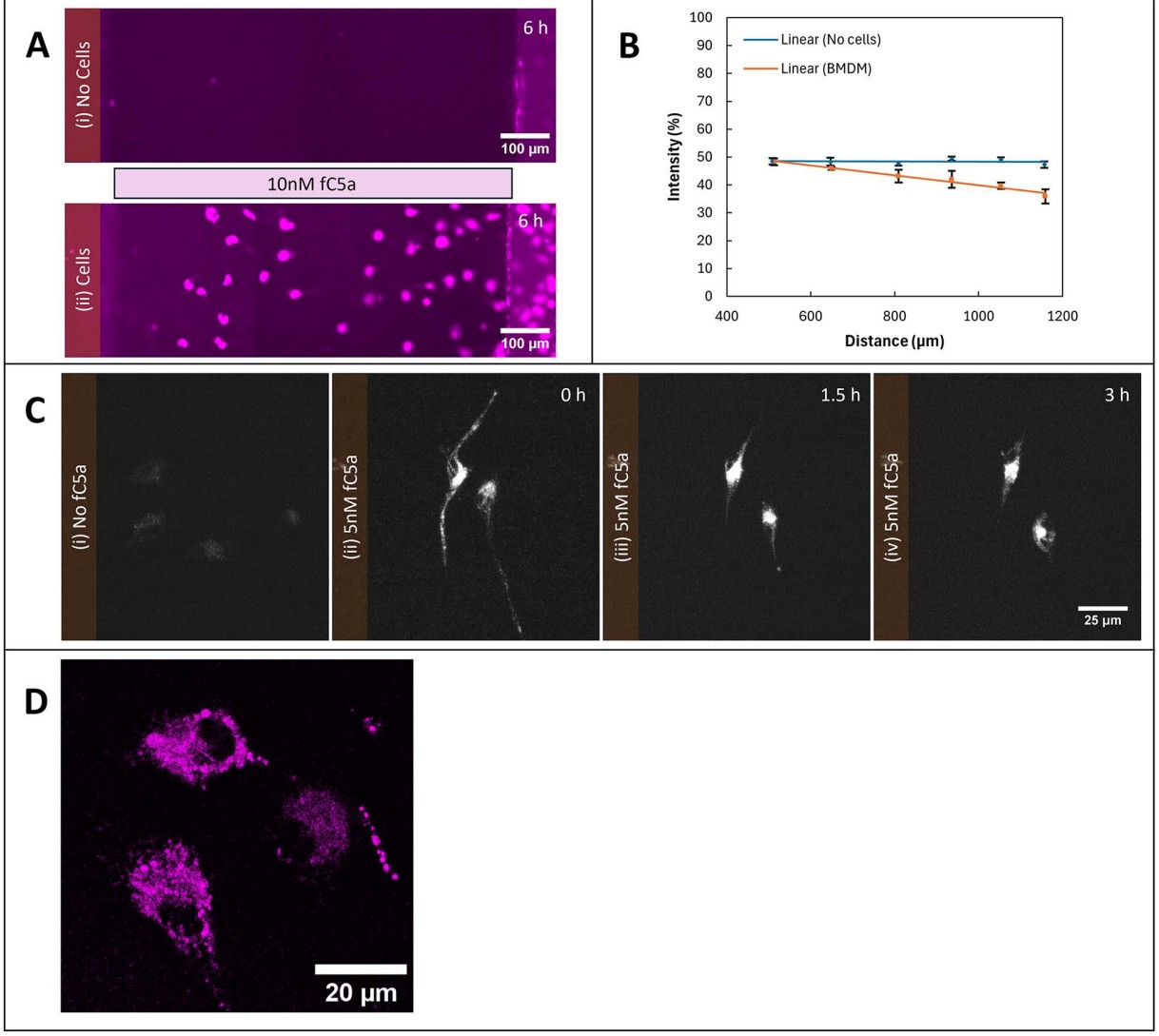

**Fig 4. Receptor-mediated uptake of C5a by BMDMs.** (A) Distribution of fluorescent C5a-AF647 after incubation without (i) or with (ii) BMDMs. (B) Quantification of fluorescence intensity along the bridge in the presence or absence of BMDMs, showing depletion of C5a near the cells. (C) Time course of C5a uptake. BMDMs were exposed to 5 nM C5a-AF647 for 1 h, washed, and imaged at 0, 1.5, and 3 h. Fluorescence localized to intracellular puncta and accumulated near the perinuclear region, consistent with endocytosis and lysosomal processing. Images were captured and processed using identical acquisition parameters. (D) Magnified view showing C5a-AF647 in vesicular structures 1.5 h after exposure. Cells prepared as in C. The data underlying this figure can be found in https://doi.org/10.5522/04/31211095.

macrophages over time. Interestingly, although both types of macrophage could perform self-generated chemotaxis, only the mouse cells were able to affect the fluorescence profile of the C5a gradient (Fig 5A and 5B), demonstrating a role of receptor-mediated endocytosis in depleting C5a from the environment. For human macrophages (Fig 5C and 5D), the gradient of fluorescence remained similar, suggesting that the C5a was mainly being inactivated by the surface bound enzyme rather than endocytosed.

The different mechanism of establishing self-generated gradients would be expected to lead to different behaviors. In particular, the $K_d$ of receptors is typically much lower than the $K_m$ of enzymes, so receptor-mediated mechanisms typically

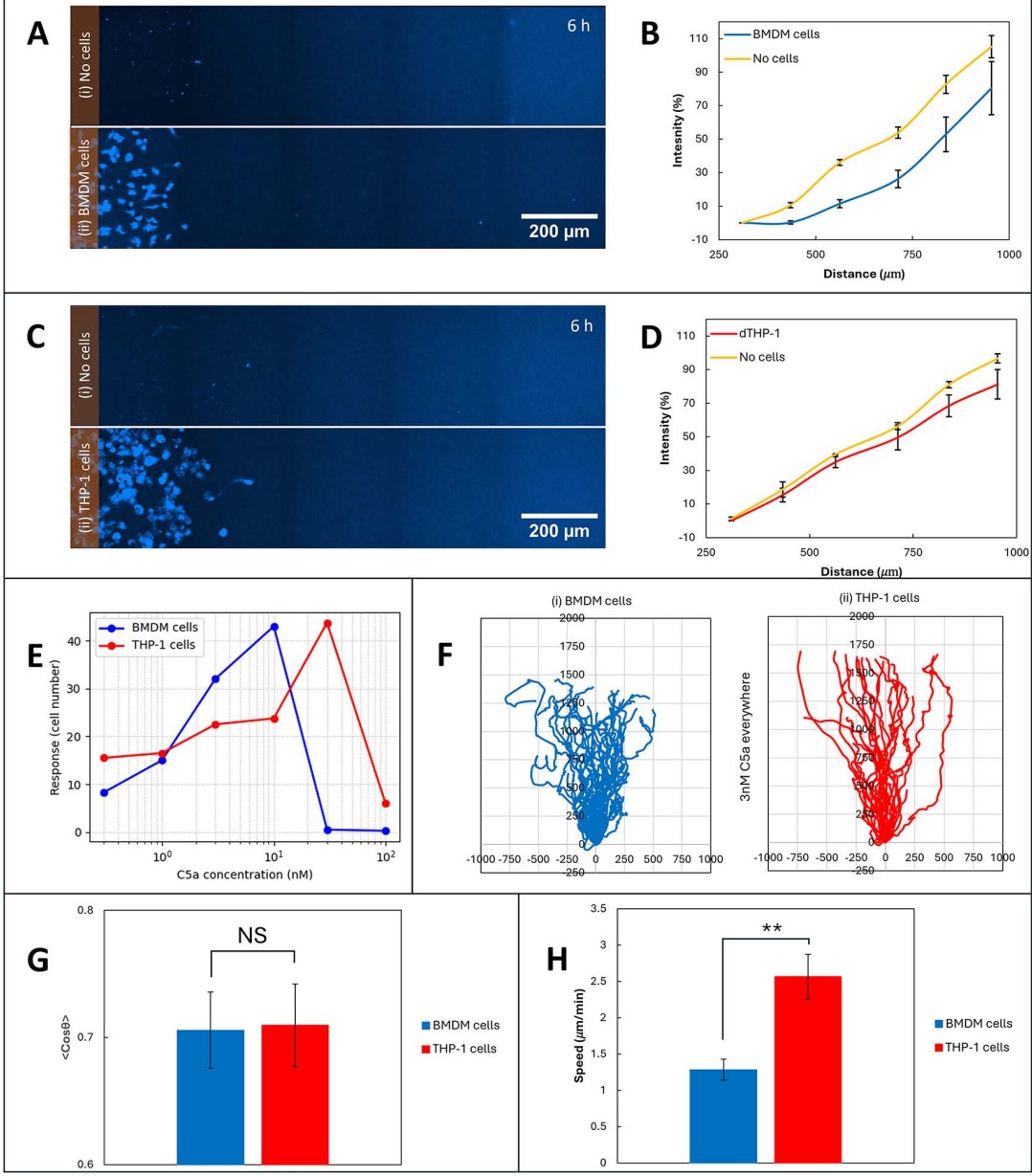

**Fig 5. Different C5a inactivation mechanisms in mouse and human macrophages. (A)** A gradient of fluorescent C5a (C5a-AF647) was established in an Insall chamber with and without mouse BMDM cells. **(B)** Measurement of fluorescent intensity along the bridge with and without BMDM cells, **(C)** Imposed gradient of fluorescent C5a with and without THP-1 cells, and **(D)** Measured intensity along the bridge with and without THP-1 cells, **(E)** Dose response of chemotaxis. The number of cells that had migrated 200 μm along the bridge was measured after 18 h for different C5a concentrations. **(F)** Spider plot of mouse and human macrophages in response to uniform 3 nM C5a. **(G, H)** Comparing the self-generated chemotaxis of mouse and human macrophages to 3 nM C5a showed an insignificant difference in their cosθ (G), but a significant difference in the mean speed (H). Bar in G and H = SE. ** represents $p$-value < 0.001. The data underlying this figure can be found in https://doi.org/10.5522/04/31211095.

work at a lower ligand concentration than enzyme-mediated ones. The dose response of both types of macrophages was assessed by counting the number of cells crossing a line 200 μm along the bridge after 18 h for various C5a concentrations. As predicted, the human cells steered most effectively at a higher C5a concentration (>30 nM versus 10 nM, Fig 5E).

Tracking the chemotaxis of both types of macrophage, using 3 nM C5a so as to be below the saturation for both types of macrophage (Fig 5F), showed that human macrophages performed much better at the higher end of the gradient. This presumably reflects their greater ability to degrade the chemoattractant in their vicinity, forming steeper and more robust gradients compared to mouse macrophages, which can only internalize one C5a molecule at a time through receptor endocytosis and are thus limited by receptor availability at the cell surface.

The collective behavior is also different—mouse macrophages show synchronized motion among themselves to form a clearly visible front wave (Fig 3A), whereas the human macrophages chemotax more independently (S7 Movie). Interestingly, the human macrophages can operate within a broader range of chemoattractant concentrations (Fig 5E), showing a clear response at very low concentrations as well as a better one at higher concentrations. This robustness is a predicted outcome of systems that use chemoattractant-degrading enzymes as well as receptor-mediated uptake [15,21].

Mouse and human macrophages behaved dissimilarly when performing self-generated chemotaxis to 3 nM of C5a (Fig 5F). Although they had a very similar time independent chemotaxis index (cosθ—see Fig 5G), the average speed of the human macrophages was twice that of mouse macrophages (Fig 5H). The speed of the chemotactic wave depends predominantly on the interaction between the Vmax of C5a degradation and its diffusion, rather than the inherent speed of the cells, so fast migration of human cells is consistent with rapid degradation of C5a, again consistent with the involvement of surface bound enzymes.

### C5aR1 mediates C5a chemotaxis

C5a is transduced by two cell surface receptors, C5aR1 and C5aR2. Most authors consider C5aR1 to be the primary chemotactic receptor [31].

C5aR2's role is less clear. It may transduce a subset of signals, or may be a scavenger receptor, in which case it could play a role in the formation of self-generated gradients. To test whether C5aR1 or C5aR2 was forming the gradient, we employed a noncompetitive receptor antagonist PMX-53, which inhibits C5a functions that act through C5aR1 but does not affect C5aR2. We verified the inhibitor was effective by imaging macrophages chemotaxing up an imposed gradient of 0–10 nM C5a in the presence or absence of 10 µM PMX-53 (Fig 6A and 6B, S8 Movie). The inhibitor is clearly effective—the untreated cells show longer and better-directed trajectories (Fig 6A, 6C) than inhibited ones (Fig 6B, 6D). The average speed of the cells with positive control and inhibitor control was 1.25 µm/min and 1.06 µm/min, respectively, implying that the inhibitor does not affect the cell's underlying ability to migrate, but mean cosθ dropped from 0.4 to 0.2 after inhibitor treatment (Fig 6E). Thus, as expected, C5aR1 is the receptor that mediates chemotaxis.

Imaging receptor-mediated uptake of C5a-AF647 in the presence and absence of PMX-53 demonstrated that C5aR1 was also the receptor mediating C5a uptake. Cells were seeded in 8-well glass bottom dishes, with or without pretreatment with 10 µM PMX-53 for 30 min. After this, 10 nM C5a-AF647 was added and incubated for 1 h, before assessing C5a uptake using a confocal microscope. As previously, untreated macrophages contained high levels of C5a, but inhibitor-treated cells showed strikingly less (Fig 6F, quantified in Fig 6G)—the total fluorescence endocytosed in the cells dropped by >60%. As receptor inhibitors are not completely efficient, this implies that an even greater fraction of the C5a uptake from the environment was mediated by C5aR1.

### Discussion

Anaphylatoxin C5a, a small peptide released as a result of complement activation, is a strong chemoattractant for macrophages [32]. Studies over many years and using widely different methods have shown mouse bone marrow-derived [33–35] and other [36,37] macrophages perform efficient chemotaxis up gradients of a range of attractants [33] including C5a [38,39]. Here we have shown, for the first time, that macrophages can perform self-generated gradient (SGG) chemotaxis towards C5a, turning a flat concentration of C5a into a gradient, then migrating up it (Fig 1B). Self-generated gradients are an efficient and often counterintuitive way for cells to probe their environment and obtain dynamic and

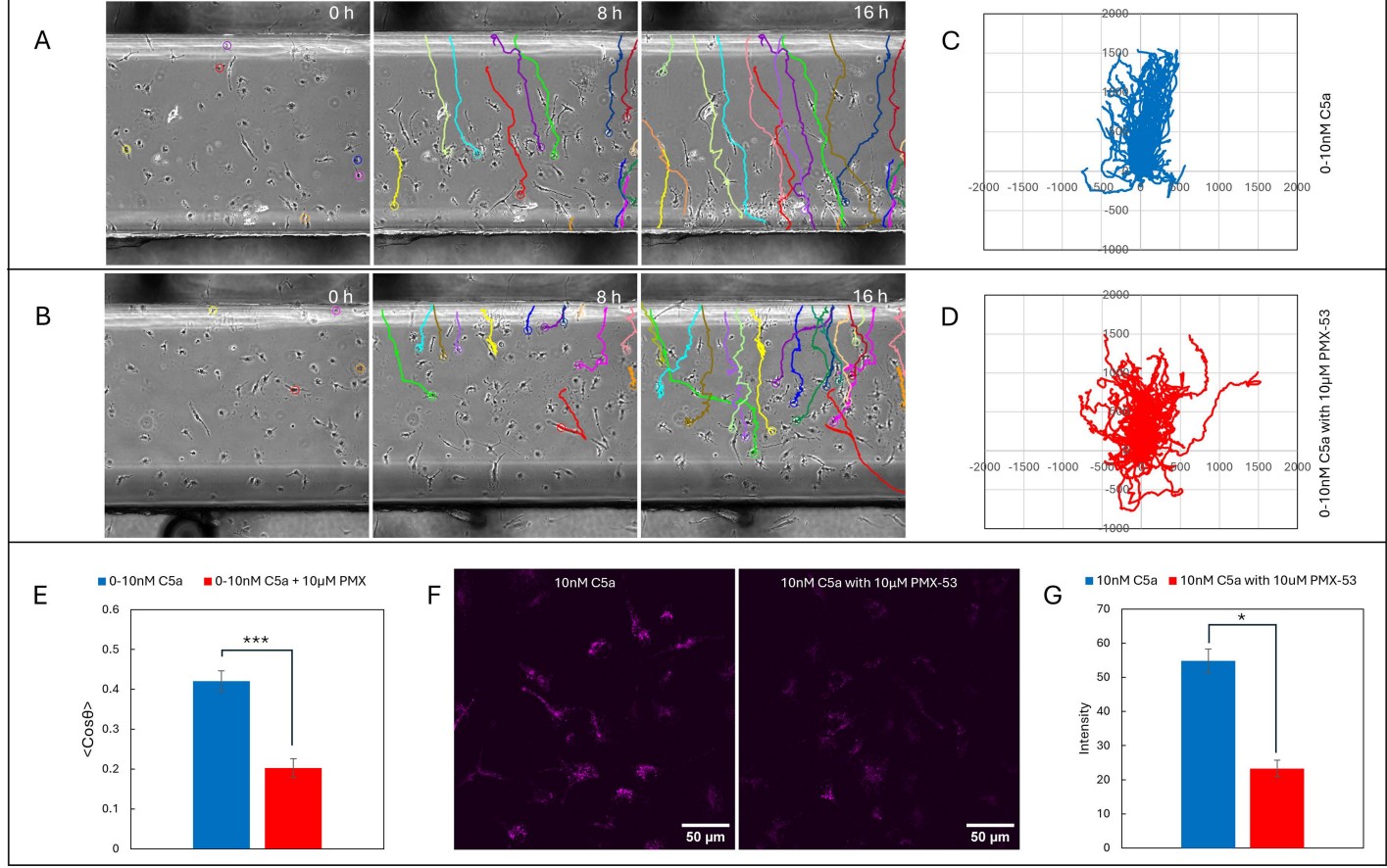

**Fig 6. Receptor-mediated C5a uptake via C5aR1.** Time-lapse images of cells subjected to **(A)** 0–10 nM C5a with cells tracks, **(B)** 0–10 nM C5a in the presence of 10 µM PMX-53 (inhibitor), **(C)** Spider plot for the 0–10 nM C5a, **(D)** Spider plot for the 0–10 nM C5a in the presence of 10 µM PMX-53, **(E)** Comparison of chemotaxis index for both the cases with and without inhibitor, **(F)** Confocal images indicating the C5a- AF647 uptake and **(G)** Quantification of cellular intensity (bars show mean± SEM, *n* = 3; ** Student *t*-test *p*-value = 0.0012). The data underlying this figure can be found in https://doi.org/10.5522/04/31211095.

complex information about the best paths [40–42]; this mode of steering is feasible and believed to occur frequently in real biological scenarios.

For efficient chemotaxis, the cells need to deplete the chemoattractant available in their immediate vicinity at a rate that is sufficient to overcome new production and diffusion from the environment. Two well-established mechanisms by which cells can degrade chemoattractants have been found to set up self-generated gradients [21,43,44]. The first uses surface bound enzymes, with their active sites on the extracellular side, that degrade and inactivate the local chemoattractant. When we first observed a front wave in macrophage chemotaxis, suggesting a self-generated gradient, membrane-bound carboxypeptidase M (CPM) offered the first candidate mechanism. CPM cleaves the C-terminal arginine of C5a to give C5a-des-Arg [27]. There are two problems with this scenario. First, CPM is only macrophage-specific in humans; mouse cells like BMDMs do not appear to express it. Second, CPM trimming of C5a does not fully inactivate it, but reduces both its receptor affinity and efficacy [45]. It is not clear whether this would of itself support a self-generated gradient. In any case, mass spectrometry of the cell supernatant exposed to cells revealed no evidence of C5a-des-Arg in mouse BMDMs. The second mechanism is through clathrin-mediated endocytosis [46], where cells take in the chemoattractant ligand

attached to its surface G-protein coupled receptor (GPCR) by endocytosing the receptor-ligand complex and targeting it to lysosomes. Our confocal experiments and their analysis indicate that BMDMs use clathrin-mediated endocytosis for C5a uptake (Fig 4).

Self-generated chemotaxis has an interesting and complex relationship with the initial concentration of the attractant. In simple SGGs the number of cells performing chemotaxis to the chemoattractant increases with the concentration of chemoattractant. There is an ideal attractant concentration. Below this, relatively few cells migrate, and the gradient is not robust. Above it, saturation makes the gradient less legible, so cells align less accurately; and it takes an increased time for the chemotaxis to start, because it takes longer for cells to remove enough attractant. This is exactly how macrophages behave in response to homogeneous C5a. We have therefore shown that cells take up C5a, that a gradient is actually formed, and cells behave as expected; it is quite clear that macrophages are fully able to use self-generated chemotaxis to C5a. It is possible that secondary attractants (another form of self-generated gradient, where cells produce self-attractants such as LTB4 [47], ATP [36], or cAMP [48] in response to chemotactic signals) are additionally involved; but the initial direction comes from the establishment of a C5a gradient.

The most surprising outcome of this research is that "normal" chemotaxis—macrophages migrating up externally-imposed, relatively steep gradients of C5a—requires the same processes as self-generated chemotaxis to work efficiently. Clearly, when cells are given a defined C5a gradient, they alter it and reinforce it to give themselves the clearest stimulus and the most information possible.

In summary, this work shows the importance of self-generated gradients in BMDM chemotaxis to C5a. In the presence of macrophages, C5a gradients are more flexible and more complex, than was previously thought.

## Materials and methods

### Cell culture

The stem cells were extracted from the bone marrow of the wild-type mice. The stem cells were supplemented with macrophage differentiation media made of DMEM, 10% heat-inactivated FBS, 1% Penicillin-streptomycin, 1% L-glutamine, 50 µM mercaptoethanol, and 25ng/ml mouse macrophage colony stimulating factor (mCSF). Fully differentiated macrophages were obtained by day 4. At day 4, the macrophages were rinsed, and fresh macrophage differentiation media were provided. The chemotaxis experiments were performed between 6 and 8 days. The cells were stimulated with 10ng/ml of LPS a day prior to the chemotaxis experiments.

### Microscopy

**Phase contrast microscopy.** The time-lapse experiments were performed using inverted Nikon ECLIPSE Ti microscope equipped with a LWD 0.52NA condenser, a motorized XY stage, and controlled by commercially available MetaMorph software. The microscope is integrated with an incubator, which enabled us to perform the experiments at 37 °C and 5% $CO_2$. For image acquisition, a 10×0.3NA air objective was used to capture the images at multiple locations with time interval of 10min.

**Confocal microscopy.** Intensity measurement experiments were performed using a Nikon A1R inverted and Zeiss 880 confocal microscope equipped with motorized stage. Where appropriate, we used commercially available (Almac Sciences (Scotland), Edinburgh) human C5a labeled with Alexa Fluor-647 on its N-terminus. Human C5a was used because of the commercial unavailability of fluorescently labeled mouse C5a. Mouse C5a and human C5a share 60% similarity in their amino acid sequence, and mouse macrophages respond and chemotax efficiently to human C5a [49]. We have also experimentally verified that BMDM cells chemotax to mouse and human C5a with comparable efficiency.

For intensity measurements relating to SGG experiments shown in Fig 4, 10nM of C5a-AF647 was provided in both inner and outer well of chemotaxis chamber with and without cells. For intensity measurement experiment for imposed gradient shown in Fig 5, 10nM C5a-AF647 was added to the inner well, either with or without cells in the outer well of

the chemotaxis chamber. For all these experiments, the assembled chambers were kept in the incubator at 37 °C and 5% $CO_2$ for 6 h, followed by the confocal imaging. Image scanning was performed with a 633 nm laser and images were acquired using either a 40×/1.3NA oil or 60×/1.4NA oil objective. Laser power was adjusted in order to obtain measurable fluorescent signal across the entire bridge. The in-built image stitching feature available in the microscope software was employed to capture the entire bridge. For image analysis, the variation of the fluorescent intensity along the bridge was measured using FIJI/ImageJ (https://imagej.net/software/fiji/).

The confocal imaging to evaluate the C5aR1 mediated uptake of C5a-AF647 (Fig 6F) was performed using a 60×/1.4NA oil objective on either a Nikon NSPARC or Zeiss 980 microscope. For analysis, regions of interest were drawn around each cell, and their fluorescent intensity was measured. The fluorescent intensity of each cell was subtracted by the average background intensity of the image, followed by measuring the average intensity of all cells in the image.

## Statistical analyses

Statistical analyses were performed using Microsoft Excel (Office 16). All experiments were independently repeated at least three times. Comparisons between groups were evaluated using two-tailed Student's *t* tests, with significance thresholds designated as $p < 0.05$ (*), $p < 0.001$ (**), and $p < 0.0001$ (***). Data are presented as mean ± standard error of the mean (SEM) unless otherwise indicated, with error bars shown for $n > 3$. For Fig 5E, variability in cell distributions between biological replicates precluded meaningful averaging; therefore, a representative dataset is shown. Although the spatial patterns differed across replicates, the overall conclusions were consistent. For Fig 1D cells were counted using the Cyto3 cell segmentation neural network retrained on 32 of our earlier macrophage images using Cellpose [50,51] combined with Ultrack [52].

## Supporting information

**S1 Fig. Self-generated gradient responses at (i) 10 nM C5a and (ii) 100 nM C5a.** The red dashed box highlights the leading cell wave. Solid black and yellow lines indicate the distance covered by cells on the bridge at low and high concentrations, respectively.
(PDF)

**S2 Fig. Effect of initial cell density on C5a dose response.** The migration response of BMDM cells seeded at (A) High cell density (6 h), (B) Medium cell density (12 h), and (C) Low cell density (24 h) for different uniform C5a dose.
(PDF)

**S3 Fig. Mass spectrometry report for C5a trimming by cultured human THP-1 macrophages incubated with hC5a.** C5a-des-R was seen under all conditions tested. Mouse BMDMs under similar conditions showed no trimming of mC5a.
(PDF)

**S1 Table. Parameters of computational model.**
(DOCX)

**S1 Movie. Self-generated C5a gradient.** Right: Mouse BMDMs migrating in constant 10 nM C5a in an Insall chamber. The top well contains BMDMs plus 10 nM C5a. Bottom well contains C5a only. Left: Negative control without C5a. Frame rate 1 frame/10 min.
(MP4)

**S2 Movie. Computational model.** Computational model reproducing the 0–10 nM imposed gradient (right) shown in Fig 1B (iii) and S4 Movie, 10 nM self-generated gradient (left) shown in Fig 1B (i) and S1 Movie, and 100 nM self-generated gradient shown in S1 Fig (ii) and S3 Movie. Parameters as shown in S1 Table. Frame rate 1 frame/10 min.
(MP4)

**S3 Movie. Complex response to different C5a doses.** Mouse BMDMs migrating in constant 10 nM and 100 nM C5a in an Insall chamber. The top well contains BMDMs plus C5a. Bottom well contains C5a only. Frame rate 1 frame/10 min.
(MP4)

**S4 Movie. Imposed C5a gradient gives similar chemotaxis to self-generated.** Mouse BMDMs migrating in an imposed gradient of 0–10 nM C5a in an Insall chamber. The top well contains only BMDMs. Bottom well contains 10 nM C5a, so a linear gradient is set up between them. Frame rate 1 frame/10 min.
(MP4)

**S5 Movie. The number of cells in a wave correlates with the amount of C5a.** Mouse BMDMs migrating in constant C5a concentrations in Insall chambers. The top well contains BMDMs plus the indicated concentration of C5a. Bottom well contains the same concentration of C5a without cells. Frame rate 1 frame/10 min.
(MP4)

**S6 Movie. Computational model for C5a dose response.** Computational model recreating the BMDMs migration in response to the different homogeneous C5a doses shown in Fig 3A & B. Frame rate 1 frame/10 min.
(MP4)

**S7 Movie. Mouse versus human cells.** Mouse BMDMs and cultured human THP-1 cells migrating in constant 3 nM C5a in an Insall chamber. The top well contains cells plus 3 nM C5a. Bottom well contains C5a only. Frame rate 1 frame/10 min.
(MP4)

**S8 Movie. Effect of C5aR1 inhibitor PMX-53.** Mouse BMDMs migrating in an imposed gradient of 0–10 nM C5a in an Insall chamber, with PMX-53 added as indicated. The top well contains only BMDMs. Bottom well contains 10 nM C5a, so a linear gradient is set up between them. Frame rate 1 frame/10 min.
(MP4)

## Author contributions

**Conceptualization:** Abhimanyu Kiran, Edward W. Roberts, Robert H. Insall.

**Formal analysis:** Abhimanyu Kiran, Robert H. Insall.

**Funding acquisition:** Edward W. Roberts, Robert H. Insall.

**Investigation:** Abhimanyu Kiran, Peter A. Thomason, David M. Versluis, Sergio Lilla.

**Methodology:** Abhimanyu Kiran, David M. Versluis, Luke Tweedy.

**Resources:** Peggy I. Paschke, Hannah Donnelly, Isabel Bravo-Ferrer, Amy Shergold, Ryan Devlin.

**Software:** Luke Tweedy, David M. Versluis.

**Supervision:** Robert H. Insall.

**Validation:** Abhimanyu Kiran, Peter A. Thomason, Hannah Donnelly, Amy Shergold, Robert H. Insall.

**Visualization:** Abhimanyu Kiran, David M. Versluis.

**Writing – original draft:** Abhimanyu Kiran, Robert H. Insall.

**Writing – review & editing:** Abhimanyu Kiran, Isabel Bravo-Ferrer, Ryan Devlin, Edward W. Roberts, Robert H. Insall.

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
