## [Editor Report · Decision Letter 0]

20 Nov 2025

Dear Robert,

Thank you for submitting your manuscript entitled "Macrophage chemotaxis steered by complex self-generated gradients of complement C5a" for consideration as a Research Article by PLOS Biology.

Your manuscript has now been evaluated by the PLOS Biology editorial staff as well as by an academic editor with relevant expertise and I am writing to let you know that we would like to send your submission out for external peer review.

Once your full submission is complete, your paper will undergo a series of checks in preparation for peer review. After your manuscript has passed the checks it will be sent out for review. To provide the metadata for your submission, please Login to Editorial Manager (https://www.editorialmanager.com/pbiology) within two working days, i.e. by Nov 24 2025 11:59PM.

Kind regards,

Ines

--

Ines Alvarez-Garcia, PhD

Senior Editor

PLOS Biology

---

## [Decision Letter · Decision Letter 1]

17 Dec 2025

Dear Robert,

Thank you for your patience while your manuscript entitled "Macrophage chemotaxis steered by complex self-generated gradients of complement C5a" was peer-reviewed at PLOS Biology. The manuscript has now been evaluated by the PLOS Biology editors, an Academic Editor with relevant expertise, and by two independent reviewers.

The reviews are attached below. As you will see, the reviewers find the conclusions interesting and worth pursuing, but they also raise a few issues that would need to be addressed before we can consider the manuscript for publication. Reviewer 1 notes that the model used for some initial results was not applied elsewhere in the paper, and that it could be useful to simulate other results. This reviewer also thinks that the ‘wave of cell migration’ described should be quantified along with the response to an initial gradient of C5a, and that it would be interesting to try to change the initial cell densities and measure responses, as well as simulating the experiments with the model. If some of these experiments are too challenging, please discuss the pertinent points and limitations in the text. Reviewer 2 mainly requests the discussion of several points, including why THP-1 cells are used rather than human peripheral blood monocyte-derived macrophages, the differences in actin cytoskeletal dynamics in mouse BMDMs vs THP-1 cells, and the potential roles of relay signals.

In light of the reviews and discussion with the Academic Editor and the rest of the team, we would like to invite you to revise the work to thoroughly address the reviewers' reports. Given the revision needed, we cannot make a decision about publication until we have seen the revised manuscript and your response to the reviewers' comments. Your revised manuscript is likely to be sent for further evaluation by all or a subset of the reviewers.

**IMPORTANT - SUBMITTING YOUR REVISION**

3. Resubmission Checklist

a) *PLOS Data Policy*

b) *Published Peer Review*

Best wishes,

Ines

--

Ines Alvarez-Garcia, PhD

Senior Editor

PLOS Biology

Reviewers' comments

Rev. 1:

The authors of this enlightening study have demonstrated that macrophages can perform self-generated gradient chemotaxis towards C5a, turning a flat concentration of C5a into a gradient, and then migrating up that gradient. They also showed that normal chemotaxis requires the same self-generated gradient. They confirmed that there is an optimal concentration of C5a. Experiments and analysis indicate that BMDMs use clathrin-mediated endocytosis for the C5a uptake.

The paper is clearly written. Here is what can improve it:

1) the model was used for the initial result in fig 1, and was not applied to anything else in the paper. It could be useful, for example, to simulate the results of fig 3.

2) The 'wave of cell migration' is just a visual impression for now. it would be better to quantify this phenomenon and report quantitative measures.

3) It would be interesting to try to change the initial cell densities and measure responses, as well as simulating such experiments with the model.

4) Quantifying a response to not only the flat concentration of C5a but also to an initial gradient of C5a would be informative.

Rev. 2:

Phagocytes are guided to sites of inflammation by complement C3a and C5a, as well as other chemoattractants. It has been known for a long time that membrane-bound enzymes can degrade extracellular chemoattractants, such as cAMP (PDE expressed in Dictyostelium) (e.g. Sucgang et al., Dev. Biol. 1997) and complement C5a (carboxypeptidase M in immune cells) (e.g. Rehli et al. 1995; Krause et al., Immunol. Rev. 1998), and thereby reshape the chemoattractant gradient. However, both enzymatic degradation and receptor-mediated internalization of chemoattractants have emerged as functionally important mechanisms of self-generated gradients (e.g. Boldajipour et al., Cell 2007; Dona et al., Nature 2013; Tweedy et al., PloS Biol. 2016; Tweedy & Insall, Front. Cell Dev. Biol. 2020), allowing chemotaxis in the absence of an imposed gradient (i.e. in a uniform concentration) or enhancing directional migration in the presence of a defined concentration gradient.

In this study, Kiran et al. nicely demonstrated that both mouse bone marrow-derived macrophages (BMDMs) and a human myeloid cell line (THP-1 cells) can self-generate a chemoattractant gradient, albeit using differents mechanisms. Using a fluorescent ligand, the authors could show that mouse BMDMs self-generate a gradient through complement C5a receptor 1 (C5aR1)-mediated complement C5a internalization. In contrast, THP-1 cells, but not BMDMs, degraded complement C5a through the activity of membrane-bound carboxypeptidase M (confirmed by mass spectrometry). Moreover, maximal cell density at the wave front of self-generated chemotaxis was observed at higher starting complement C5a concentrations, which could be explained by the higher rate of chemoattractant depletion by enzymatic activity compared to receptor-mediated internalization and receptor recycling. That is, mouse BMDMs, which lack the enzyme, are limited by receptor availability. The authors additionally showed, using BMDMs, that pretreatment with a Toll-like receptor 4 ligand, lipopolysacchardies (LPS), augmented chemotaxis.

The experiments are well designed and the data nicely support the conclusions. This study advances the concept of self-generated gradients as a key element in chemotactic signaling and offers a framework to both interpret and predict the behavior of motile immune cells.

Comments

1. The authors should discuss why THP-1 cells (human monocytic leukemia line) were used instead of, for example, human peripheral blood monocyte-derived macrophages.

2. Please discuss the differences in actin cytoskeletal dynamics in mouse BMDMs versus THP-1 cells. The former appear to have low rear end RhoA/actomyosin activity, whereas the latter exhibit a rounded morphology.

3. The authors should briefly discuss the potential roles of relay signals, such as auto- and paracrine Leukotriene B4 (LTB4) and ATP signaling in self-generated gradients and chemotaxis.

Minor comments

1. New Figure 1: It would be helpful to include a new figure (Figure 1), which shows the Insall chamber in a schematic form. This will help explain the lateral gray-shaded structures in some of the figures and supplementary movies, e.g. Fig. 1C, and save the "busy reader" from looking up the original description of the chamber.

2. Abstract: The word "enzymatic" is perhaps superfluous in the text "through carboxypeptidase-mediated enzymatic degradation"?

3. Figure 1C: It would help the reader if "Self-generated gradient" was included in the label at the end of end panel. Otherwise, the "busy reader" needs to consult the legend.

4. The Methods section contains passages where the grammar and phrasing could be refined. Examples are highlighted below.

5. Methods: Cell culture - "The cells were stimulated with 10ng/ml of LPS experiment a day prior to the chemotaxis experiments" should be "The cells were stimulated with 10ng/ml of LPS [delete "experiment"] a day prior to the chemotaxis experiments" or similar.

6. Methods: Confocal microscopy - "Intensity measurement experiments were performed using Nikon A1R inverted and Zeiss confocal microscope equipped with motorized stage" should be "Intensity measurement experiments were performed using a [add "a"] Nikon A1R inverted and Zeiss confocal microscope equipped with motorized stage".

7. Methods: Confocal microscopy - "AlexaFluor-647®" should be "Alexa Fluor-647®"

8. Methods: Confocal microscopy - "Mouse and human C5as" should be "Mouse C5a and human C5a"

9. Methods: Confocal microscopy - "For intensity measurement experiment for SGG experiments shown in Figure 4" should be "For intensity measurements relating to SGG experiments shown in Figure 4" or similar.

10. Methods: Confocal microscopy - "10nM C5a-AF647® was added in the inner well with and without cells in the outer well of the chemotaxis chamber" should be "10nM C5a-AF647® was added to the inner well, either with or without cells in the outer well of the chemotaxis chamber" or similar ["to" instead of "in", and ", either with or without"].

11. Methods: Confocal microscopy - "Image scanning was done with 633nm laser and images were acquired using 40X/1.3NA oil and 60X/1.4NA oil objective" should be "Image scanning was done with "a" 633nm laser and images were acquired using "either" a 40X/1.3NA oil or 60X/1.4NA oil objective".

12. Methods: Confocal microscopy - "The confocal imaging to evaluate the C5aR1 mediated uptake of C5a AF647® (Figure 6F) were performed using a 60x/1.4NA oil objective on a Nikon NSPARC and Zeiss 980" should be The confocal imaging to evaluate the C5aR1 mediated uptake of C5a-AF647® (Figure 6F) were performed using a 60x/1.4NA oil objective on "either" a Nikon NSPARC "or" Zeiss 980 "microscope""or similar.

13. Be consistent with C5a-AF647® - it also appears as C5a-AF647 or C5a AF647® in the manuscript.

14. Methods: Confocal microscopy - "For analysis, outline was made around all the cells, and their fluorescent intensity was measured" should be "For analysis, regions of interest were drawn around each cell, and their fluorescent intensity was measured" or similar.

15. Fig. 6 Legend: "student's t-test" should be "Student's t-test" [capital "S" and "t" in italics]

16. Legend of Movie 7: "10 µM PMX-53 everywhere" does not sound very scientific. Perhaps "10 µM PMX-53 (uniform throughout)" would be better.

17. The figure legends lack a consistent format. Some use "(A) … (B) …", while others do not. Please revise all figure legends to follow the journal's standardized style and ensure consistency throughout.

---

## [Decision Letter · Decision Letter 2]

27 Feb 2026

Dear Robert,

Thank you for your patience while we considered your revised manuscript entitled "Macrophage chemotaxis steered by complex self-generated gradients of complement C5a" for publication as a Research Article at PLOS Biology. This revised version of your manuscript has been evaluated by the PLOS Biology editors, the Academic Editor and one of the original reviewers.

Based on the review, we are likely to accept this manuscript for publication, provided you satisfactorily address the data and other policy-related requests stated below my signature.

In addition, we would like you to consider a suggestion to improve the title:

"Macrophages can self-generate complex gradients and enhance chemotaxis towards complement component 5a"

We expect to receive your revised manuscript within two weeks.

*Published Peer Review History*

*Press*

Best wishes,

Ines

--

Ines Alvarez-Garcia, PhD

Senior Editor

PLOS Biology

DATA POLICY:

Fig. 1D; Fig. 3C; Fig. 4B; Fig. 5B, D-H and Fig. 6C, D, E, G

CODE POLICY

Per journal policy, if you have generated any custom code during the course of this investigation, please make it available without restrictions. Please ensure that the code is sufficiently well documented and reusable, and that your Data Statement in the Editorial Manager submission system accurately describes where your code can be found. More information on our Code Policy, what and how to share can be found here: https://journals.plos.org/plosbiology/s/code-availability

Reviewers' comments

Rev. 1:

The authors addressed all comments well.

---

## [Editor Report · Decision Letter 3]

13 Mar 2026

Dear Robert,

Thank you for the submission of your revised Research Article entitled "Macrophages Self-Generate And Refine Chemotactic Gradients During Migration Towards Complement C5a" for publication in PLOS Biology. On behalf of my colleagues and the Academic Editor, Sui Huang, I am delighted to let you know that we can in principle accept your manuscript for publication, provided you address any remaining formatting and reporting issues. These will be detailed in an email you should receive within 2-3 business days from our colleagues in the journal operations team; no action is required from you until then. Please note that we will not be able to formally accept your manuscript and schedule it for publication until you have completed any requested changes.

PRESS

Sincerely,

Ines

--

Ines Alvarez-Garcia, PhD

Senior Editor

PLOS Biology
